# Effect of Different Intracanal Medicaments on the Viability and Survival of Dental Pulp Stem Cells

**DOI:** 10.3390/jpm12040575

**Published:** 2022-04-04

**Authors:** Shilpa Bhandi, Shankargouda Patil, Nezar Boreak, Hitesh Chohan, Abdulaziz S. AbuMelha, Mazen F. Alkahtany, Khalid H. Almadi, Thilla Sekar Vinothkumar, A. Thirumal Raj, Luca Testarelli

**Affiliations:** 1Department of Restorative Dental Sciences, College of Dentistry, Jazan University, Jazan 45412, Saudi Arabia; shilpa.bhandi@gmail.com (S.B.); nboraak@jazanu.edu.sa (N.B.); drhiteshchohan@yahoo.co.in (H.C.); vinothkumar_ts@yahoo.com (T.S.V.); 2Department of Maxillofacial Surgery and Diagnostic Sciences, Division of Oral Pathology, College of Dentistry, Jazan University, Jazan 45412, Saudi Arabia; 3Department of Restorative Dental Sciences, College of Dentistry, King Khalid University , Abha 61421, Saudi Arabia; aabumelha@kku.edu.sa; 4Department of Restorative Dental Science, Division of Endodontics, College of Dentistry, King Saud University, Riyadh 11362, Saudi Arabia; malkahtany@ksu.edu.sa (M.F.A.); kalmadi@ksu.edu.sa (K.H.A.); 5Department of Conservative Dentistry and Endodontics, Saveetha Dental College, Saveetha Institute of Medical and Technical Sciences, Saveetha University, Chennai 600077, India; 6Department of Oral Pathology and Microbiology, Sri Venkateswara Dental College and Hospital, Chennai 600130, India; thirumalraj666@gmail.com; 7Department of Oral and Maxillo-Facial Sciences, Università di Roma La Sapienza, 00185 Roma, Italy; luca.testarelli@uniroma1.it

**Keywords:** calcium hydroxide, dental pulp, doxycycline, endodontic regeneration, root canal medicaments, stem cells

## Abstract

Background: Stem cells play an important role in the success of regenerative endodontic procedures. They are affected by the presence of medicaments that are used before the induction of bleeding or the creation of a scaffold for endodontic regeneration. This study examines the effects of different intracanal medicaments on the viability and survival of dental pulp stem cells at different doses and over different exposure times. Methods: Dental pulp stem cells were cultured from healthy third molar teeth using the long-term explant culture method and characterized using flow cytometry and exposed to different concentrations of calcium hydroxide, doxycycline, potassium iodide, triamcinolone, and glutaraldehyde, each ranging from 0 (control) to 1000 µg/mL. Exposure times were 6, 24, and 48 h. Cell viability was measured using the MTT assay, and apoptosis was measured using the Annexin V-binding assay. Results: All medicaments significantly reduced cell viability at different concentrations over different exposure times. Calcium hydroxide and triamcinolone favored cell viability at higher concentrations during all exposure times compared to other medicaments. The apoptosis assay showed a significant increase in cell death on exposure to doxycycline, potassium iodide, and glutaraldehyde. Conclusion: The intracanal medicaments examined in our study affected the viability of dental pulp stem cells in a time and dose-dependent manner. They also adversely affected the survival of dental pulp stem cells. Further studies are needed to better understand the effect of prolonged exposure to medicaments according to clinical protocols and their effect on the stemness of dental pulp stem cells.

## 1. Introduction

Dental caries is a highly prevalent disease in the human population and a major contributor to the global burden of oral disease [1]. It can cause inflammation and necrosis of the pulpal and radicular tissues. Conventional treatment for an infected root canal weakens the tooth and may be ineffective in removing microbes from the root canal [2,3]. Regenerative endodontic procedures (REPs), a potential alternative, are endorsed and described by endodontic specialist associations [4,5]. The clinical procedures for regenerative endodontics involve: minimal instrumentation, disinfecting the root canal, applying an intracanal medicament, provoking bleeding into the canal, and capping followed by a proper coronal seal [4]. The criteria for a successful outcome are based on clinical and radiographic signs up to 24 months. However, histologically, the outcome is unpredictable [6]. The long-term success of the procedure is also uncertain. A recent systematic review showed that REPs fail due to persistent infections, and the failure is usually detected after a year [7].

Regenerative endodontic procedures rely on the principles of tissue engineering and take advantage of the regenerative abilities of stem cells [8]. Dental pulp stem cells (DPSCs), described by Gronthos et al., are mesenchymal stem cells found in the dental pulp of human teeth [9]. They can differentiate into multiple lineages, making them valuable for stem cell therapy [8,9,10]. Microbes may adversely influence the proliferation and regenerative potential of stem cells [11]. Since REPs advise the minimal instrumentation of root canals, they are more dependent on intracanal medicaments with antimicrobial activity to provide a sterile environment for pulpal regeneration. The recommended medicament for the disinfection of root canals is calcium hydroxide [5]. Studies show that triple antibiotic paste and double antibiotic paste are more effective antimicrobials for the disinfection of root canals compared to calcium hydroxide [12,13].

Intracanal medicaments also affect the survival and proliferation of mesenchymal stem cells. A study that used clinical concentrations of triple antibiotic paste and double antibiotic paste found that they are detrimental to the survival of stem cells in the apical papilla [14]. This study also indicated that calcium hydroxide benefits these stem cells. Antibiotics also affect the proliferation and attachment of DPSCs [15]. The cytotoxicity of these antimicrobial drugs has also been found dependent on the time of exposure and the concentration of the drug used [16]. Mesenchymal stem cells in the oral regions, including DPSCs, that are crucial to the regenerative process are exposed to intracanal medicaments. The aim of this study, therefore, was to examine the effects of different intracanal medicaments on the survival and viability of DPSCs in a time and concentration-dependent manner.

## 2. Materials and Methods

### 2.1. Sample Collection

Human premolar teeth indicated for orthodontic extraction in 17 to 18-year-old healthy individuals were the source for the DPSCs. It was ensured that the extracted teeth were healthy, without any dental caries or periodontal disease. According to institutional ethics guidelines, the study was approved by Scientific Research, College of Dentistry, Jazan University (CODJU-19682), and informed permission was acquired. The pulp was removed under sterile circumstances and delivered directly to the laboratory for further processing.

### 2.2. Culture and Expansion of Human Dental Pulp Stem Cells

Dental Pulp Stem Cells (DPSCs) were isolated and characterized utilizing the explant culture approach, described in detail in a previous publication [17]. Concisely, the pulp tissue was shredded into tiny bits and placed in 35-mm polystyrene plastic culture dishes. The tissues were soaked in enough Fetal Bovine Serum (FBS) (Gibco, Rockville, MD, USA) to completely cover them (small drops approx. 10 μL). For explant tissue, including FBS, a 24-h incubation at 37 °C and 5% CO_2_ was performed; the entire DPSC culture system was then maintained in DMEM (Invitrogen, Carlsbad, CA, USA) supplemented with 20% FBS and antibiotic–antimycotic solution at the same temperature and CO_2_ settings. The culture medium was replaced twice weekly, and an inverted phase–contrast microscope was used to examine cell growth, health, and morphology. Cells were detached at 70–80% confluence and moved to a larger 25-cm^2^ polystyrene culture flask using 0.25 percent Trypsin-EDTA solution (Invitrogen, Carlsbad, CA, USA and Nunc, Rochester, NY, USA). Confluent DPSCs were separated using a 0.25 percent Trypsin-EDTA solution and then passaged in for expansion and subsequent tests. In this study, cells from passage 4 were employed.

### 2.3. Characterization of DPSCs with Flow Cytometry

Confluent DPSCs were harvested with trypsinization and washed twice with PBS for a cell surface marker analysis. Anti-human-CD73-APC, anti-human-CD90-APC, anti-human-CD105-APC, anti-human-CD34-PE, anti-human-CD45-FITC, and anti-human-HLA-DR-APC antibodies (Miltenyi Biotec, Bergisch Gladbach, Germany) were then added to the cells and incubated for 30 min at 4 °C. Antibody-labeled cells were washed twice in PBS before being counted on an Attune NxT flow cytometer (Thermo Fisher Scientific, Waltham, MA, USA) at a rate of 20,000 cells per sample. To identify and distinguish between positive and negative signals, isotype controls were used.

### 2.4. Treatment Groups

For the MTT assay, calcium hydroxide (Calh), doxycycline (Doxy), potassium iodide (Pot), triamcinolone (Tri), and glutaraldehyde (Glut) (Sigma Aldrich, St. Louis, MO, USA) were used. Each medicament was studied at concentrations of 0 µg/mL (control), 10 µg/mL, 25 µg/mL, 50 µg/mL, 100 µg/mL, 250 µg/mL, 500 µg/mL, and 1000 µg/mL. The stem cells were exposed to the medicaments for 6 h, 24 h, and 48 h. For the apoptosis assay, selected concentrations of Calh at 25 µg/mL, Doxy at 10 µg/mL, Pot at 10 µg/mL, Tri at 25 µg/mL, and Glut at 25 µg/mL were used to treat DPSCs for 24 h.

### 2.5. 3-(4,5-Dimethylthiazol-2-yl)-2,5-diphenyltetrazolium Bromide (MTT) Assay of DPSC following Quercetin Treatment

The cells were treated after being seeded in 96-well plates (1 × 10^4^ cells per well). The MTT assay was used to assess the drug cytotoxicity in DPSCs. The cells were seeded in 96-well plates and cultured for 6, 24, and 48 h with the appropriate medium. After mixing, plates were incubated for 4 h at 37 °C, and MTT solution (Sigma-Aldrich Corp., St. Louis, MO, USA) at a concentration of 0.5 mg/mL was added to each well. The medium was then withdrawn, and each well was filled with 100-µL dimethyl sulfoxide (DMSO) (Sigma-Aldrich Corp., St. Louis, MO, USA). A Multiskan Spectrum spectrophotometer (Thermo Scientific, San Jose, CA, USA) was used to detect the absorbance at 570 nm.

### 2.6. Analysis of Cell Apoptosis Using Annexin V-FITC/PI Assay

Annexin V-FITC/PI (BD Pharmingen, San Diego, CA, USA) staining was used in the apoptosis detection experiment. DPSCs were planted at a density of 1 × 10^5^ cells per well in 12-well plates and incubated for 24 h. In addition, the cells were given suitable medication concentrations mixed with full media (DMEM + 10% FBS) and cultured for 48 h. The cells were collected and stained with Annexin V-FITC reagent, both treated and untreated. In the absence of light, the cells were incubated for 15 min at room temperature. After incubation, the cells were exposed to PI reagent, and flow cytometry was used to determine the percentage of apoptotic cells.

### 2.7. Statistical Analysis

The data were presented as the mean and standard deviation of three independent experimental values. The data were analyzed using GraphPad Prism 8 software (GraphPad Software, La Jolla, CA, USA) to compare all of the experimental groups using an unpaired *t*-test (two-tailed), and *p* < 0.05 was taken as significant (* *p* < 0.05 and ** *p* < 0.01).

## 3. Results

### 3.1. Characterization of Human Dental Pulp Stem Cells (DPSCs)

Dental pulp stem cells were isolated from healthy human teeth showed typical mesenchymal stem cell (MSC)-like morphology (Figure 1A). They also had MSC-specific cell surface markers. The DPSCs showed more than 85% positive cells for CD73, CD90, and CD105 (Figure 1B–E). They showed negative expression for MHC class II antigen HLA-DR and hematopoietic markers CD34 and CD45 (Figure 1F).

### 3.2. Cytotoxicity of Intracanal Medicaments

The medicaments showed variable cytotoxicity at different concentrations and time points (Figure 2A–O). Calcium hydroxide showed a significant reduction in cell viability at 50 µg/mL or higher after exposure for 24 and 48 h. The cytotoxicity was highly significant when exposed to 250 µg/mL or more. After 6 h, there was a significant reduction in cell viability at exposure to 500 µg/mL of calcium hydroxide.

Doxycycline was significantly cytotoxic at 6 h of exposure to 500 µg/mL. For a 24-h exposure, 25 µg/mL and above significantly reduced the cell viability. However, for 48-h exposure, cytotoxicity was at 10 µg/mL and higher. Potassium iodide reduced the cell viability significantly at 250 µg/mL after 6 h of exposure, 50 µg/mL after 24-h exposure, and 100 µg/mL after 48-hour exposure. Triamcinolone was significantly cytotoxic at 250 µg/mL after 6 h, 100 µg/mL after 24 h, and 25 µg/mL after 48 h. Glutaraldehyde significantly reduced cell viability at 500 µg/mL after 6 h and 50 µg/mL after 24 h and 48 h.

### 3.3. Apoptosis Assay

The Annexin V-FITC/PI assay showed similar survival between controls and those treated with calcium hydroxide and triamcinolone (Figure 3A–F). A comparison of the percentage of apoptosis of DPSCs when exposed to different medicaments showed significantly higher apoptosis with doxycycline at 10 µg/mL, potassium iodide at 10 µg/mL, and glutaraldehyde at 25 µg/mL after 24 h (Figure 3G–N).

## 4. Discussion

This study examined the effects of different antimicrobial intracanal medicaments at various concentrations on the viability and survival of DPSCs over different periods. The results indicated that DPSCs showed better viability with lower concentrations of all medicaments for 24 or 48 h, but the medicaments were tolerated at higher concentrations when exposed for 6 h. Both concentration and time of exposure of intracanal medicaments affected the viability of DPSCs. The apoptosis assay indicated that DPSCs showed a significantly higher percentage of cell death with doxycycline, potassium iodide, and glutaraldehyde compared to the controls.

The long-term explant method for culturing DPSCs is a simple and inexpensive method of culturing DPSCs, especially when limited resources are available. The cultured DPSCs showed characteristic mesenchymal stem cell (MSC) markers and the absence of typical non-MSC markers (Figure 1A–F). The cells obtained by this method have shown characteristics similar to those obtained by the enzymatic method [17]. They also show the ability to differentiate into different lineages similar to MSCs and the potential for differentiation demonstrated by DPSCs [9,18,19].

Calcium hydroxide is a recommended intracanal medicament for disinfection during regenerative endodontic procedures [20]. The exposure of DPSCs to calcium hydroxide induces their proliferation, osteogenic differentiation, and mineralization via the mitogen-activated protein kinase pathway [21]. Calcium hydroxide has an antimicrobial effect for 48 h; after which, it declines [22]. In our study, calcium hydroxide exposure up to 25 µg/mL did not significantly affect the viability of dental pulp stem cells after 48 h. In contrast, a previous study showed that calcium hydroxide is not toxic for stem cells of the apical papilla (SCAPs) and increases cell survival and proliferation [23]. However, an evaluation of calcium hydroxide’s cytotoxicity towards SCAPs at its minimum bactericidal concentration showed higher toxicity than antimicrobial combinations [24]. Our study presents a similar result, additionally highlighting that both concentration and time of exposure contribute to its cytotoxic effect. After 6 h, concentrations as high as 250 µg/mL of calcium hydroxide were well-tolerated by DPSCs. However, on exposures of 24 and 48 h, DPSCs were adversely affected by one-tenth the concentration.

Previous studies found doxycycline to be a good intracanal antimicrobial irrigant and medicament [25,26,27]. However, the concentrations of doxycycline used in these studies were much higher than the cytotoxic concentrations determined in our research. Doxycycline caused significant cytotoxicity after 24 h at 25 µg/mL and after 48 h at 10 µg/mL. These were the lowest concentrations among all the studied medicaments. However, 500 µg/mL of doxycycline was required to significantly reduce the viability of DPSCs after 6 h. Its better tolerance to short exposure favors its use as an intracanal irrigant. However, the concentration must be kept low due to its substantivity of up to 3 weeks after 10 min of exposure [26]. It also induced the highest percentage of apoptosis in DPSCs among all medicaments compared in the study. Thus, doxycycline alone seems unfavorable for regenerative procedures, especially when DPSCs play a major role in regeneration.

Potassium iodide has been used as an antimicrobial and anticaries agent [28,29]. It is an effective disinfectant irrigant solution [30]. We found that potassium iodide was better tolerated than doxycycline. The cytotoxic concentration was similar to calcium hydroxide after 24 h, but it was better tolerated than calcium hydroxide for 48 h. However, it induced significantly higher apoptosis than the controls. Potassium iodide and doxycycline were tested at their lowest concentrations for the apoptosis assay (10 µg/mL). Both medicaments induced significantly higher cell death.

Triamcinolone, a corticosteroid, has been used with tetracycline in the form of ledermix paste as an initial dressing for patients with endodontic symptoms for direct or indirect pulp capping [31]. It showed a significant adverse effect on DPSC survival above a concentration of 100 µg/mL after 24 h, which was higher than calcium hydroxide. However, it reduced to 25 µg/mL after 48 h, which was less than calcium hydroxide and potassium iodide but greater than doxycycline. Similarly, a previous study that examined the impact of corticosteroids found a dose-dependent effect on the viability of mesenchymal stem cells [32]. Our study indicated an additional time-dependent effect. Both triamcinolone and potassium iodide showed similar cytotoxic concentrations after 6 h, lower than calcium hydroxide.

Glutaraldehyde has been used as an irrigant and an intracanal medicament [33,34]. It has also been used as a medicament following pulpotomy [35]. It showed significant cytotoxic effects on DPSCs at a concentration of 50 µg/mL after exposure for 24 or 48 h and 500 µg/mL after 6 h, similar to calcium hydroxide. However, glutaraldehyde caused significantly higher apoptosis in DPSCs.

It is important to determine which medicaments can be used safely and effectively in regenerative procedures without stem cells losing their viability and regenerative potential. The viability of DPSCs with intracanal medicaments seems time and concentration-dependent. A limitation of this study is that the study focused on the effects of the medicaments up to 48 h. Current clinical guidelines recommend leaving the medicament in place for 1–4 weeks [4]. The concentrations of the medicaments were increased by greater amounts as the dose was increased. Further studies that examine the effect of medicaments with fixed increases in the concentration can help determine the ideal concentration of the medicaments. From the results obtained in this study, concentrations of the medicaments that did not decrease cell survival after 48 h can be further tested for prolonged exposure and according to the suggested clinical protocols. These medicaments also need to be studied concerning their disinfection capability. The minimum concentration required should not affect the survival of DPSCs. Residual medicaments, their substantivity, and its implications on the scaffold for endodontic regeneration also need examination. In our study, calcium hydroxide and triamcinolone were tolerated best. While triamcinolone can be combined with an antimicrobial, its systemic absorption can be a concern.

Regenerative endodontic procedures involve a complex interplay between stem cells and scaffolds. Recently, platelet concentrates have been used as aids for wound healing after extractions and are effective in controlling postoperative bleeding [36]. The ability of platelet-rich concentrates to serve as reservoirs of growth factors has been used as an advantage for regenerative endodontics as well. They have shown promise as scaffolds for endodontics, providing outcomes comparable to a conventional blood clot [37]. In addition, a recent systematic review examined in vitro studies that explored the antimicrobial potential of platelet-rich concentrates and found them potentially effective against oral and periodontal microbes [38]. However, further studies are needed to examine if the platelet concentrates are effective antimicrobials for endodontic regeneration given the limited quantity and exposure in a root canal. Even when intracanal medicaments are removed, there may be a small concentration remaining in the dentinal tubules. Therefore, the effect of intracanal medicaments on the regenerative and antimicrobial activity of scaffolds such as platelet concentrates, the stem cell secretome, and their differentiation into various lineages also need to be studied further.

## 5. Conclusions

In this study, we tested the effects of different intracanal medicaments on the viability and survival of dental pulp stem cells when exposed to different concentrations for 6, 24, and 48 h. We found that calcium hydroxide and triamcinolone were the least likely to adversely affect cell viability and survival among all the medicaments tested. Doxycycline induced the highest percentage of apoptosis in the stem cells. Further studies are needed to test the antimicrobial activity of calcium hydroxide and triamcinolone either alone or in combination with other antimicrobial medicaments. There is also a need to focus on the long-term effects of these medicaments on stem cells due to remnants in the dentinal tubules after the medicament is removed from the root canal.

## Figures and Tables

**Figure 1 jpm-12-00575-f001:**
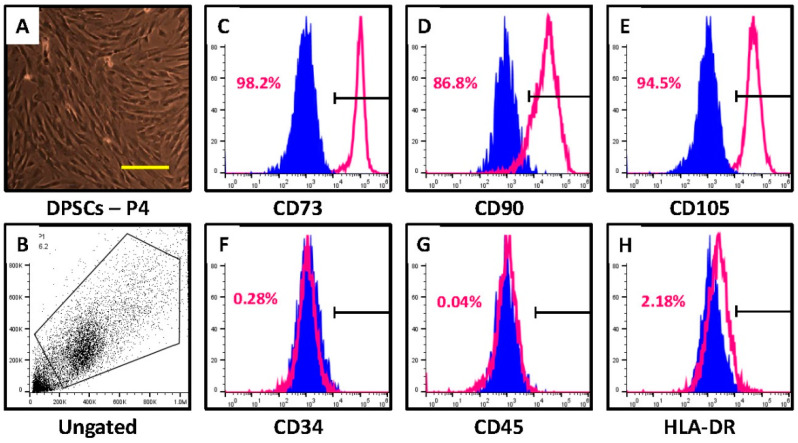
Characterization of DPSCs for mesenchymal stem cell properties. (**A**) Photomicrograph of DPSCs at passage 4. Scale bar = 100 μm. (**B**–**H**) DPSCs were checked for MSC-specific positive markers CD73, CD90, and CD105 and MSC-specific negative markers CD34, CD45, and HLA-DR.

**Figure 2 jpm-12-00575-f002:**
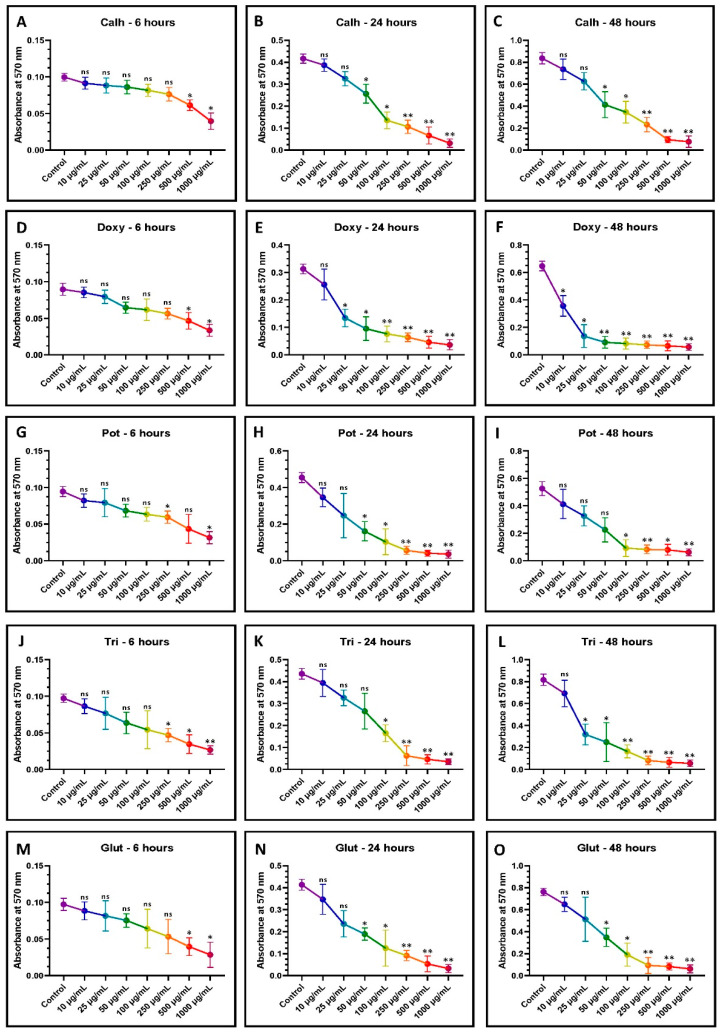
MTT assay for the assessment of cytotoxicity. (**A**–**O**) DPSCs were treated with various concentrations of individual drugs (0 µg/mL, 10 µg/mL, 25 µg/mL, 50 µg/mL, 100 µg/mL, 250 µg/mL, 50 µg/mL, and 1000 µg/mL) for 6, 24, and 72 h, and a comparative analysis was done to check the cytotoxicity of the drugs to DPSCs. ns is not significant; * *p* < 0.05 and ** *p* < 0.01.

**Figure 3 jpm-12-00575-f003:**
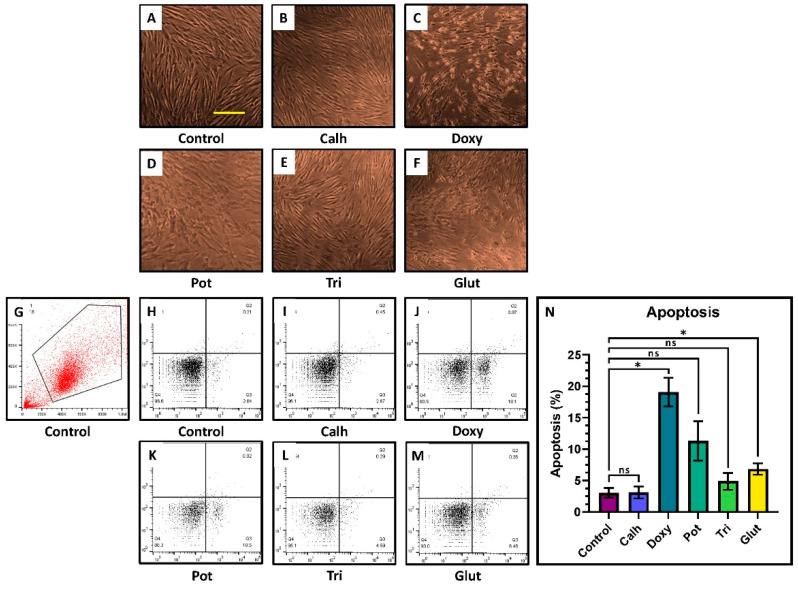
Morphologic and apoptosis analysis in DPSCs treated with selected concentrations of drugs. (**A**–**F**) Morphology of DPSCs after treatment with selected concentrations of drugs. (**G**–**N**) Comparative analysis of apoptosis by annexin V assay. ns is not significant; * *p* < 0.05.

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
