# Peer review of "Effect of Different Intracanal Medicaments on the Viability and Survival of Dental Pulp Stem Cells"

_jpm, 2022, doi:10.3390/jpm12040575_

Round 1
Reviewer 1 Report
Why did the authors decide to extract third molars from patients aged 17-18 years?
The authors did not mention the inclusion and exclusion criteria for the patients in the sutdy.
What does ” enough Fetal Bovine Serum” mean? How the authots could quantify this?
At line 119 the authors mention twice 50 µg/mL ” 50 µg/mL, 100 µg/mL, 250 µg/mL, 50 µg/mL”.
Figure 2 and fig. 3 are blurry and the graphics are very little.
At line 210 the authors speak about previous studies but they mention just one study.
Please also mention the limitations of the study.
Author Response
Reviewer 1:
- Why did the authors decide to extract third molars from patients aged 17-18 years?
Response: These were patients who were referred to the oral surgery department for extraction of pre-molars as part of the orthodontic treatment. By mistake it was written as third molars. It is now corrected in the revised draft.
- The authors did not mention the inclusion and exclusion criteria for the patients in the study.
Response: Following was added: Human pre-molar tooth indicated for orthodontic extraction in 17–18-year-old healthy individuals were the source for the DPSCs. It was ensured that the extracted tooth was healthy without any dental caries or periodontal disease.
- What does ” enough Fetal Bovine Serum” mean? How the authots could quantify this?
Response: FBS was introduced to the tissues with micropipette and approximately 10 μL quantity was used.
- At line 119 the authors mention twice 50 µg/mL ” 50 µg/mL, 100 µg/mL, 250 µg/mL, 50 µg/mL”.
Response: This was changed to 50 µg/mL, 100 µg/mL, 250 µg/mL, 500 µg/mL.
- Figure 2 and fig. 3 are blurry and the graphics are very little.
Response: Replaced all the figures with sharper images.
- At line 210 the authors speak about previous studies but they mention just one study.
Response: Corrected
- Please also mention the limitations of the study.
Response: Limitations of the study were mentioned in lines 255 – 262.
Reviewer 2:
Title: Effect of different intracanal medicaments on the viability and survival of dental pulp stem cells
- What is the main question addressed by the research?
To assess the effect of different intracanal medicaments on the viability and survival of dental pulp stem cells at different doses and over different exposure times.
- Is it relevant and interesting?
The article is relevant and interesting.
- How original is the topic?
The topic is current.
- What does it add to the subject area compared with other published material?
The authors have collected and analyzed a great deal of new data.
- Is the paper well written?
Yes, the article is well written.
- Is the text clear and easy to read?
Yes, but minor English editing is required.
- Are the conclusions consistent with the evidence and arguments presented?
Yes, the conclusions consistent with the evidence and arguments presented but further studies are necessary to confirm authors’ hypothesis.
- Do they address the main question posed?
Yes, the Authors addressed the main question posed.
Other comments:
- English language: Minor English editing is required.
Response: The articles was checked and errors rectified.
- Introduction: This section needs some improvements. Please refer to the use of platelet concentrates as a scaffold for endodontic regeneration, and I also would suggest inserting a sentence on academic debate on platelet concentrate effect: <<Efficacy of platelet concentrates in promoting wound healing and tissue regeneration is at the center of a recent academic debate [PMID: 31116189]>>.
Response: We thank the reviewer for the insightful comment. We felt that the use of platelet rich concentrates would be better suited to the discussion section as it provides an area for further exploratory research as well.
- Materials and Methods: This section was properly prepared
- Results: This section was properly prepared
- Discussion: What is the main theme that emerges from the authors' analysis? Is the study design a limitation? Please improve.
Response: The main theme that emerged from this study was the effect of time and dose dependent effect of intracanal medicaments on the viability of dental pulp stem cells. Thi has been highlighted in lines 194-195. The limitations of the study have been mentioned in lines 255 – 262.
- Conclusion: This section was properly prepared but further studies are necessary to confirm authors’ hypothesis.
Thanks for the opportunity to review this manuscript.
Reviewer 3:
Congratulations to the authors on the idea. There are some editorial errors in the paper (see comments). In terms of content seems to be correct.
1# L201 - "17" - Is this a mistake?
Response: Corrected
2# L203 and L282 please add a period at the end of the sentence.
Response: Corrected
3# References are not written in jpm style. Please correct the whole thing.
Response: Corrected
https://www.mdpi.com/journal/jpm/instructions
Reviewer 2 Report
Manuscript ID: jpm-1670451
Title: Effect of different intracanal medicaments on the viability and survival of dental pulp stem cells
1.What is the main question addressed by the research?
To assess the effect of different intracanal medicaments on the viability and survival of dental pulp stem cells at different doses and over different exposure times.
2.Is it relevant and interesting?
The article is relevant and interesting.
3.How original is the topic?
The topic is current.
4.What does it add to the subject area compared with other published material?
The authors have collected and analyzed a great deal of new data.
5.Is the paper well written?
Yes, the article is well written.
6.Is the text clear and easy to read?
Yes, but minor English editing is required.
7.Are the conclusions consistent with the evidence and arguments presented?
Yes, the conclusions consistent with the evidence and arguments presented but further studies are necessary to confirm authors’ hypothesis.
8.Do they address the main question posed?
Yes, the Authors addressed the main question posed.
Other comments:
- English language: Minor English editing is required.
- Introduction: This section needs some improvements. Please refer to the use of platelet concentrates as a scaffold for endodontic regeneration, and I also would suggest inserting a sentence on academic debate on platelet concentrate effect: <<Efficacy of platelet concentrates in promoting wound healing and tissue regeneration is at the center of a recent academic debate [PMID: 31116189]>>.
- Materials and Methods: This section was properly prepared
- Results: This section was properly prepared
- Discussion: What is the main theme that emerges from the authors' analysis? Is the study design a limitation? Please improve.
- Conclusion: This section was properly prepared but further studies are necessary to confirm authors’ hypothesis.
Thanks for the opportunity to review this manuscript.
Author Response

(The authors gave the same response as above.)

Reviewer 3 Report
Congratulations to the authors on the idea. There are some editorial errors in the paper (see comments). In terms of content seems to be correct.
1# L201 - "17" - Is this a mistake?
2# L203 and L282 please add a period at the end of the sentence.
3# References are not written in jpm style. Please correct the whole thing.
https://www.mdpi.com/journal/jpm/instructions
Author Response

(The authors gave the same response as above.)
